

# A data science challenge for converting airborne remote sensing data into ecological information

Sergio Marconi[1], Sarah J. Graves[2], Dihong Gong[3],
Morteza Shahriari Nia[3], Marion Le Bras[4], Bonnie J. Dorr[4],
Peter Fontana[4], Justin Gearhart[1], Craig Greenberg[4], Dave J. Harris[5],
Sugumar Arvind Kumar[3], Agarwal Nishant[3], Joshi Prarabdh[3],
Sundeep U. Rege[3], Stephanie Ann Bohlman[2], Ethan P. White[5] and
Daisy Zhe Wang[3]

[1] School of Natural Resources and Environment, University of Florida, Gainesville, FL, USA
[2] School of Forest Resources and Conservation, University of Florida, Gainesville, FL, USA
[3] Department of Computer and Information Science and Engineering, University of Florida, Gainesville, FL, USA
[4] National Institute of Standards and Technology, Gaithersburg, MD, USA
[5] Department of Wildlife Ecology and Conservation, University of Florida, Gainesville, FL, USA

Corresponding authors
Stephanie Ann Bohlman,
sbohlman@ufl.edu
Ethan P. White, ethanwhite@ufl.edu
Daisy Zhe Wang,
daisyw@cise.ufl.edu

## ABSTRACT

Ecology has reached the point where data science competitions, in which multiple groups solve the same problem using the same data by different methods, will be productive for advancing quantitative methods for tasks such as species identification from remote sensing images. We ran a competition to help improve three tasks that are central to converting images into information on individual trees: (1) crown segmentation, for identifying the location and size of individual trees; (2) alignment, to match ground truthed trees with remote sensing; and (3) species classification of individual trees. Six teams (composed of 16 individual participants) submitted predictions for one or more tasks. The crown segmentation task proved to be the most challenging, with the highest-performing algorithm yielding only 34% overlap between remotely sensed crowns and the ground truthed trees. However, most algorithms performed better on large trees. For the alignment task, an algorithm based on minimizing the difference, in terms of both position and tree size, between ground truthed and remotely sensed crowns yielded a perfect alignment. In hindsight, this task was over simplified by only including targeted trees instead of all possible remotely sensed crowns. Several algorithms performed well for species classification, with the highest-performing algorithm correctly classifying 92% of individuals and performing well on both common and rare species. Comparisons of results across algorithms provided a number of insights for improving the overall accuracy in extracting ecological information from remote sensing. Our experience suggests that this kind of competition can benefit methods development in ecology and biology more broadly.

## INTRODUCTION

In many areas of science and technology there are tasks for which solutions can be optimized using well-defined measures of success. For example, in the field of image analysis, the goal is to accurately characterize the largest proportion of images (*Solomon & Breckon, 2011*). When a clear measure of success can be defined, one approach to rapidly improving the methods used by the field is through open competitions (*Carpenter, 2011*). In these competitions, many different groups attempt to solve the same problem with the same data. This standardization of data and evaluation allows many different approaches to be assessed quickly and compared. Because the problems are well defined and data is cleaned and organized centrally, competitions can allow involvement by diverse participants, from those with domain expertise, to those in fields like modeling and machine learning.

In fields outside of ecology, these competitions have yielded rapid advances in the accuracy of many tasks. One well-known example of this is the ImageNET image classification competition (*Krizhevsky, Sutskever & Hinton, 2012*). For the past 5 years, teams have competed in classifying 100,000s of images that has resulted in a major increase in classification accuracy from only 70% in 2010 to 97% in 2017. This success has resulted in the rapid growth of competitions for solving common data science problems through both isolated competitions and major platforms like Kaggle (https://www.kaggle.com/). Kaggle has run over 200 competitions ranging from industry challenges predicting sales prices of homes, to scientific questions like detecting lung cancer from lung scans. In general, life and environmental sciences, including ecology, have only recently begun to recognize the potential value of competitions. A few ecology-related competitions have been run recently, including competitions quantifying deforestation in the Amazon basin (https://www.kaggle.com/c/planet-understanding-the-amazon-from-space) and counting sea lions in Alaska (https://www.kaggle.com/c/noaa-fisheries-steller-sea-lion-population-count). However, these are far from common and, as a result, most ecologists are unaware of, and have had few opportunities to participate in, data science competitions.

In recent years, ecology has reached the point where these kinds of competitions could be productive. Large amounts of open data are increasingly available (*Reichman, Jones & Schildhauer, 2011*; *Hampton et al., 2013*; *Michener, 2015*) and areas of shared interest around which to center competitions are increasingly prominent. One of these shared areas of interest is converting remote sensing data into information on vegetation diversity, structure and function (*Pettorelli et al., 2014*, *2017*; *Eddy et al., 2017*). We ran a competition to improve three tasks that are central to converting airborne remote sensing (images and vertical structure measurements collected from airplanes) into the kinds of vegetation diversity and structure information traditionally collected by ecologists in the field: (1) crown segmentation, for identifying the location and size of individual trees (*Zhen, Quackenbush & Zhang, 2016*); (2) alignment to match ground truth data on trees with remote sensing data (*Graves et al., 2018*); and (3) species classification to identify trees to species (*Fassnacht et al., 2016*). If these three tasks can be conducted

with a high degree of accuracy, it will allow scientists to map species locations over large areas, and use them to understand the factors governing the individual level behavior of natural systems at scales thousands of times larger than possible from traditional field work (*Barbosa & Asner, 2017*).

To create this competition, we used data from the National Ecological Observatory Network (NEON; *Keller et al., 2008*) funded by the U.S. National Science Foundation (NSF). NEON collects data from a wide range of ecological systems following standardized protocols. One of the core sets of observations comes from the Airborne Observation Platform (AOP) that collects high resolution LiDAR and hyperspectral images across ~10,000 ha for dozens of sites across the US (http://www.neonscience.org). NEON also collects associated data on the vegetation structure at each site, which supports the building and testing of remote sensing based models. In addition to providing the openly available data needed for this competition, NEON also provides an ideal case for competitions because the methods are standardized across sites and data collection will be conducted at dozens of locations annually for the next 30 years. This means that the methodological improvements identified by the competition can be directly applied to hundreds of thousands of hectares of remotely sensed images and continual improvements can be made by regularly rerunning the competition. As a result, this competition has the potential to produce major gains in the quality of the ecological information that can be extracted from this massive data collection effort.

In addition to producing important improvements for NEON remote sensing products, this competition should also broadly benefit efforts to convert airborne remote sensing into ecological information. A major challenge in current assessments of airborne remote sensing tasks is determining whether published assessments of different methods generalize to the broad application of the methods as a whole, or are specific to the particular dataset and evaluation metrics being used. While this is a general problem for method comparison, it is particularly acute in many areas of remote sensing because: (1) most papers do not compare their methods to other approaches; (2) when comparisons are made it is typically between a new method and a single alternative; (3) different papers focus on different datasets; and (4) different papers often use different evaluation metrics and fail to specifically identify the best evaluation metric for a given task. *Zhen, Quackenbush & Zhang (2016)* have highlighted the importance of changing this culture to produce extensive method comparisons using consistent data and evaluation metrics to drive the field of crown segmentation forward. By design, competitions provide single core datasets and consistent evaluation metrics to allow direct comparisons among many different approaches.

To capitalize on the benefits of competitions for overcoming barriers of comparing methods and determining how well different approaches to common data science task generalize, the National Institute of Standards and Technology (NIST) has been developing a data science evaluation series (DSE, https://www.nist.gov/itl/iad/mig/data-science-evaluation). This program has developed methodologies for evaluating progress in data science research through iterative examination of a range of problems, with the goal of devising a general evaluation paradigm to address data science problems

**Table 1 Summary of methods used by the participants.**

| Task1 | | |
|---|---|---|
| **Group** | **Method** | **Reference** |
| FEM | itcSegment | *Dalponte, Frizzera & Gianelle (2018)* |
| Shawn | Watershed based on CHM and NDVI | *Taylor (2018)* |
| Conor | Watershed based on CHM and NDVI | *McMahon (2018)* |
| **Task2** | | |
| FEM | Euclidian distance of spatial coordinates, height, and crown radius | *Dalponte, Frizzera & Gianelle (2018)* |
| Conor | RMS minimization of relations between geographic coordinates and estimated crown diameter | *McMahon (2018)* |
| **Task3** | | |
| FEM | Support vector machine | *Dalponte, Frizzera & Gianelle (2018)* |
| Conor | Ensemble of maximum likelihood classifiers based on structural and spectral features | *McMahon (2018)* |
| StanfordCCB | Gradient boosting and random forest ensemble | *Anderson (2018)* |
| GatorSense | Multiple instance adaptive cosine estimator (MI-ACE) | *Zou, Gader & Zare (2018)* |
| BRG | Multilayer perceptron neural network | *Sumsion et al. (2018)* |

**Note:**
Methods used by each participant are grouped by tasks, and briefly described. For more details of each algorithm, see the references in column 3.

that span diverse disciplines, domains, and tasks. As a part of the early stages of DSE, a pilot evaluation was run using traffic data, which was then followed by this competition on converting remote sensing data to information on trees. As a component of this endeavor, NIST researchers identified general classes of data science problems (*Dorr et al., 2015*, *2016a*, *2016b*; *Greenberg et al., 2014*) and produced a framework for evaluating methods both within an individual domain (like in this paper) and across domains (e.g., allowing algorithms for similar tasks to be applied to both traffic and ecological problems). The NIST DSE platform was used as the foundation for the NIST DSE Plant Identification with NEON Remote Sensing Data (https://www.ecodse.org) DSE. The organization of the datasets, development of the tasks, data descriptions, evaluation metrics design, submission formats, participation information, and rules all fell within the NIST–DSE framework.

Here, we present the details of the initial run of this data science competition for converting remote sensing to data on individual trees. We present the details of the tasks and data, and summarize and synthesize the results from the participants. In a set of short accompanying papers and preprints the participants describe the methods used (Table 1), present detailed results for those methods, and discuss lessons learned and future directions for these methods (*Anderson, 2018*; *Dalponte, Frizzera & Gianelle, 2018*; *Taylor, 2018*; *McMahon, 2018*; *Sumsion et al., 2018*; *Zou, Gader & Zare, 2018*). Finally, we discuss the broad potential for competitions in ecology and the biological sciences more generally.
**Table 2 Data products and sources (*National Ecological Observatory Network, 2016*).**

| Name | NEON data product ID | Data date | How it was used |
|---|---|---|---|
| Woody plant vegetation structure | NEON.DP1.10098 | 2015 | Task 2 vegetation structure |
| Spectrometer orthorectified surface directional reflectance—flightline | NEON.DP1.30008 | 2014 | Task 1, 2, and 3 RS data (Hyperspectral) |
| Ecosystem structure | NEON.DP3.30015 | 2014 | Task 1, 2, and 3 RS data (Canopy height model) |
| High-resolution orthorectified camera imagery | NEON.DP1.30010 | 2014 | Task 1, 2, and 3 RS data (RGB photos) |
| Field ITC | Internal | 2017 | Task 1 ITC data; Task 3 to extract pixels per each crown |

**Note:**
Information about data products can be found on the NEON data products catalogue (http://data.neonscience.org/data-product-catalog).

## MATERIALS AND METHODS

### NEON data

We used NEON–AOP data (from year 2014) and field collected data (from years 2015 to 2017) for the Ordway-Swisher Biological Station (Domain D03, OSBS) in north-central Florida. The NEON field data was from 43 permanently established plots across the OSBS site, which are stratified across three land cover types (*Homer et al., 2015*). The field measurements were the NEON vegetation structure data that provides information on the stem location, taxonomic species, stem size, tree height, and in some cases two measurements of crown radius (Table 2).

Four NEON–AOP remote sensing data products were used; LiDAR point cloud data, LiDAR canopy height model (CHM), hyperspectral surface reflectance, and high resolution visible color (RGB) photographs (Table 2). The LiDAR point cloud data provide information about the vertical structure of the canopy. Data consists of a list of spatial 3D coordinates, with an average resolution of four to six points per square meter. The CHM data provides one m spatial resolution information on the spatial variation in canopy height. Hyperspectral data provides surface reflectance from 350 to 2,500 nm at one m spatial resolution and allows development of spectral signatures to identify object categories. The RGB photographs provide 0.25 m spatial resolution information in the visible wavelengths. The higher spatial resolution relative to the other data products may be helpful to separate trees that are close to one another and to refine tree crown boundaries. The RGB data was the only data type not available for all plots (39 out of 43 total). NEON provides geographically registered files of these data products across the entire NEON site. The data was clipped to 80 × 80 m subsets to capture the full 40 × 40 m field plot with a 20 m buffer on each side. The buffer was used to include any trees with their base in the plot but with a crown that fell outside of the NEON plot boundary.

## Individual tree crown field mapping data

Generating field-validated individual tree crowns (ITCs) required spatially matching individual trees measured in the field to the remote sensing image of their crowns taken from above the canopy. The ITCs were generated in the field on a tablet computer and GIS software. This process was done after both NEON remote sensing and field data had been acquired and processed. First, the 2014 NEON images were loaded in a GIS application on a tablet computer that was connected to an external GPS device. The GIS application displayed the GPS location and the NEON digital images. Second, NEON plots were visited between September 2016 and April 2017. A field-technicians from our team located all tree crown that fell within a NEON plot and had branches that were in the upper canopy and visible in the NEON image. Third, with the aid of the GPS location, and the technicians' skills in visual image analysis, the crown boundaries of individual trees were digitized in the GIS application. As the crown delineation was done 2 years after image collection, the technician did not include any crowns that could not be definitively linked to crowns in the image, such as trees that had fallen or had severe crown damage. While the LiDAR and RGB data was used to aid in tree crown delineation, the ITC polygons were made in reference to the hyperspectral data. This is important to consider when there is geographic misalignment among the three data products. The result of the field mapping process was spatially explicit polygon objects that delineated the crown boundaries of individual trees. These polygons were linked to field data by the NEON identification number, or field-based species identification.

## Train test split

Training data for the segmentation task consisted of a subset of 30 out of 43 plots (~70%). The ITCs were provided as ground truth to allow participants to apply supervised methods. Plots were selected to have a consistent 0.7 to 0.3 training-testing ratio both in number of plots, and number of ITCs (Table 3). The splitting resulted in a training dataset of 452 out of 613 ITCs. Since the OSBS NEON site is characterized by three different ecosystem types, we split the data accordingly to ensure each ecosystem was split in the 0.7 to 0.3 ratio. Separate polygon files were provided for each NEON plot. All ITC files had a variable number of polygons, and each polygon represented a single tree. LiDAR and hyperspectral derived data was made available to participants for all tasks. The RGB data were provided only when available. For the alignment task, we used only data from individual trees shared by the vegetation structure and the ITCs, resulting in a total of 130 entries. We split data in a 0.7 to 0.3 training-test ratio, following the same rationale described for segmentation. For the classification task we used data from all ITC crowns. Again, data were split in a 0.7 to 0.3 ratio. In this case, we stratified training-test samples by species labels (e.g., *Pinus palustris*, *Quercus laevis*). As a result, around 70% of the trees for each species belonged to the training set, the other 30% to the test set. We grouped species whose occurrences were less than four into a general category labelled as "Other," because their individual numbers were considered too

**Table 3 Overview of train-test data split by task and ecosystem type.**

|  | Task 1 | | Task 2 | | Task 3 | |
|---|---|---|---|---|---|---|
|  | **Plots** | **ITC** | **Plots** | **ITC** | **Plots** | **ITC** |
| **Train** | | | | | | |
| EF | 22 | 349 | 17 | 82 | 22 | 349 |
| EHW | 2 | 52 | 0 | 0 | 2 | 52 |
| WWET | 6 | 9 | 1 | 2 | 6 | 9 |
| Total | 30 | 452 | 19 | 84 | 30 | 452 |
| **Test** | | | | | | |
| EF | 9 | 144 | 7 | 28 | 9 | 144 |
| EHW | 1 | 21 | 0 | 0 | 1 | 21 |
| WWET | 3 | 7 | 1 | 2 | 3 | 7 |
| Total | 13 | 172 | 8 | 30 | 13 | 172 |

Notes:

The columns present respectively the number of NEON plots (Plots) and individual tree crowns (ITC) provided per task and ecosystem type.

EF, evergreen forest; EHW, emergent herbaceous wetland; WWET, woody wetland.

few to allow any learning. We consider the "Other" category potentially useful to discriminate rare, undefined species from the rest of the dataset.

## Timeline and participants

The DSE was announced one month in advance of making the data available (September 1, 2017), and participants were allowed to register until the final submission date (December 15, 2017). Participants could work on any or all of the tasks. There were two submission deadlines, with the first deadline providing an opportunity to get feedback on a submission evaluated on the test data before the final submission. A total of 84 groups showed interest in participating, 14 formally registered, and six teams submitted results. Teams came from a number of institutions including teams from outside the United States. The six teams were: (1) BYU, a team composed of four researchers from the Bioinformatics Research Group (BRG); (2) Conor, a team from University of Texas at Austin composed of a single researcher; (3) FEM, a team composed of three researchers of the Fondazione Edmund Mach (Italy); (4) GatorSense, a team composed of five members, all affiliated to University of Florida (but not involved in organizing the competition); (5) Shawn, a team composed of a single researcher at University of Florida; and (6) StanfordCCB, a single researcher affiliated with Stanford University.

## Competition tasks

### Segmentation

The crown segmentation task aims to determine the boundaries of tree crowns in an image. While image segmentation is a well developed field in computer science (*Badrinarayanan, Kendall & Cipolla, 2017*; *Mane, Kulkarni & Randive, 2014*), delineating tree crowns in a forest is a particularly complex task (*Ke & Quackenbush, 2011*; *Bunting & Lucas, 2006*). Most of the complexity is driven by the fact that individual crowns overlap, look similar to each other, and can show different shapes depending on the
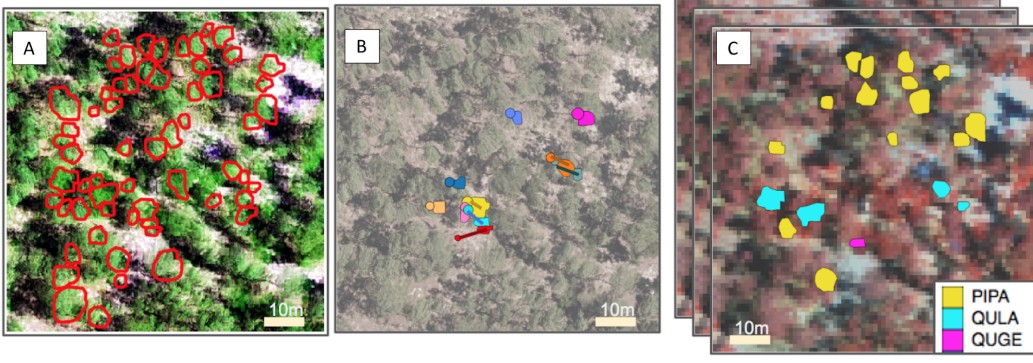

**Figure 1 Representation of the pipeline for the three competition tasks.** From left to right, (A) For the segmentation task, crowns (ITCs) delineated in the field (red); the background is a composite of the hyperspectral data (R = 829 nm, G = 669 nm, B = 473 nm) overlaid by LiDAR CHM. (B) For the alignment task, stem locations scaled by stem diameter (circles) and field ITCs (irregular polygons) overlaid on a desaturated RGB image. Both ITCs and stem locations are colored by stem identity. Lines indicating the offset between crowns and stems. (C) For the classification task, field ITCs colored by species code. The background is a composite of the hyperspectral data (R = 829 nm, G = 549 nm, B = 473 nm). Superimposed images are a graphical representation of the stack of 426 raster bands. Species shown in the legend are: *Pinus palustris* (PIPA), *Quercus laevis* (QULA), and *Quercus geminata* (QUGE).

environment and developmental stage (*Duncanson et al., 2014*). The spatial resolutions of the NEON hyperspectral and CHM LiDAR derived data (1 m²) are also relatively low compared to crown sizes. In addition, these data are also different than most image data in that they have very high spectral resolution, which may facilitate the task of distinguishing neighboring tree crowns especially if coupled to LiDAR data. As a result of these complexities, there is no widely agreed upon solution to the crown segmentation problem, as described in *Zhen, Quackenbush & Zhang (2016)*. Different classes of algorithms perform best in different ecoregions, or even within a single forest. For example, the same method can perform well in an open canopy area and poorly in a closed canopy portion of the same stand.

For the segmentation task we asked participants to delineate tree crowns in the 80 × 80 m field-plot area using remote sensing data and the ITC polygons collected in the field (Fig. 1). For a more detailed state of the art review, we point the reader to *Zhen, Quackenbush & Zhang (2016)*.

*Performance metric.* We used the mean pairwise Jaccard coefficient, J(A, B), as the performance metric for the segmentation task (*Real & Vargas, 1996*). The J(A, B) is a measure of similarity and diversity between pairs of objects, and is formulated as:

$$(A, B) = \frac{|A \cap B|}{|A \cup B|} = \frac{|A \cap B|}{|A| + |B| - |A \cap B|}$$

Where *A* and *B* are respectively the observed and predicted ITCs. By definition, the J(A, B) is a value between 0 and 1, where 0 stands for no overlap, and 1 for a perfect match.

The score for the segmentation task is the average of the plot-level scores for each pair of crowns; that is, the average J(A, B) calculated on every measured ITC with the single

most overlapping predicted crown. We used the Hungarian algorithm (*Kuhn, 1955*) to match predicted and ground truth crowns. The Hungarian algorithm is an optimization method used to pair labels from two different groups to one another, given that the same label cannot be used twice. For this task, it was used to determine which was the best assignment between each singular field ITC and a unique predicted ITC. We chose this method because it is simple to interpret, does not require assignment of predicted crowns to specific ITCs by the participants, and provides a continuous measure. We penalized cases where predicted polygons overlapped with each other by disregarding the intersecting area in the numerator of the Jaccard coefficient.

Although it was not an official scoring criterion, we also analyzed the confusion matrix of their predictions to detect how the errors were distributed. The confusion matrix is a table where predicted and ground truth labels are represented by columns and rows, respectively. In the context of crown delineation, labels are true positive, false positive, and true negative for each of the pixels. Given this information, we could determine and aggregate the number of false/true positives and negatives.

*Algorithms.* Our baseline prediction consisted of applying the Chan–Vese watershed algorithm (*Chan & Vese, 2001*), which consists of a geometric active contour model whose growing function is based on the negative of the 1 m$^2$ resolution CHM. Polygons boundaries were drawn by applying a segmentation mask to each predicted crown and following their pixels' perimeter. We chose this baseline because it is a documented, simple, and widely applied method with room for improvement. Three groups participated in the segmentation task and each applied a different algorithm. The Conor group applied a three step method that first filtered pixels based on an greenness threshold (based on NDVI, the normalized difference vegetation index), then extracted local maxima from the CHM using a linear moving window, and finally ran a watershed segmentation seeded by the local maxima (*McMahon, 2018*). The FEM group applied a growing region algorithm based on relative distance and difference in reflectance between neighbor pixels (*Dalponte et al., 2015*; *Dalponte, Frizzera & Gianelle, 2018*). For this purpose, they used the hyperspectral images, and tuned the method by visual analysis on the training set. The Shawn group used a watershed algorithm on the CHM, filtering the scene by NDVI threshold (*Taylor, 2018*).

## Alignment

Once crown location, position, and shape are recognized, it is important to accurately identify which objects in the images are linked to the data collected on the ground. Although both remote sensing and field data collection are georeferenced, these data products use different methods for geolocation. Moreover, field data coordinates locate the central stem (trunk) position, instead of the crown's centroid, which can be offset from each other, especially in closed-canopy forests. The differences in stem and crown location could lead to substantial misalignment between the two products, and consequently to misattributed information that could affect the quality of further inference. This task is known as alignment and is the second step of the pipeline. The goal of alignment is to correctly label each tree crown polygon to a single tree in the ground data, thus allowing

data collected on the ground (e.g., species identity, height, stem diameter, tree health) to be accurately associated with remote sensing data. For this round, we envisioned the alignment task as a 1:1 labelling problem (Fig. 1). We provided ITC data for crowns sampled in the field only and asked participants to link each single ITC to a specific field label. We acknowledge that this is an oversimplification of the real problem because each single ground label could be potentially confused with several apparent crowns in proximity that were not included in the field-mapped ITC dataset.

*Performance metric.* Performance of matching field stem locations to ITCs was evaluated using the trace of the prediction matrix divided by the sum over the values in that matrix. This method was chosen based on the following reasoning. In the testing stage, suppose we have a set of remotely sensed data (ITC) denoted as $\{p_n | n = 1, \ldots, N\}$, and ground truth data denoted as $\{g_n | n = 1, \ldots, N\}$. We know in advance that there is a unique one-to-one mapping between the P and G sets. Without loss of generality, assume $p_n$ should be mapped to $g_n$ for $n = 1, \ldots, N$. For each data point $p_i$, a program predicts a non-negative confidence score that should be aligned with ground truth data point $ij$, which forms a prediction matrix $M = (m_{i,j})$ where $i,j = 1, \ldots, N$. Then, the quality of prediction can be measured by the following scoring function:

$$\text{Score} = \frac{\text{trace}(M)}{\Sigma_{i,j} m_{i,j}}$$

where trace $(\cdot)$ represents trace of a matrix and $M$ represents the prediction matrix which has been aligned in the order which matches the ground truth.

*Algorithms.* Our baseline prediction was the application of naive Euclidean distance from the stem location to the centroid of the ITC. We chose this method because of its simplicity and margins for improvements. Two groups participated in this task and applied different algorithms. Both were based on the Euclidean distance between field stem and each of the ITCs included in the dataset. Euclidean distance was calculated by using East and North UTM spatial coordinates, as well as crown height and radius. The groups calculated these values using allometric relationships whenever tree height and crown size were missing from the field data. The Conor group used crown diameter as a measure of tree size (*McMahon, 2018*). Euclidean distances were adjusted for the average plot-level offset in the training data to compensate for location biases consistently within a plot. The FEM group applied the Euclidean distance based on spatial coordinates, tree height, and crown radius (*Dalponte, Frizzera & Gianelle, 2018*). FEM used an allometric equation to estimate the crown radius from tree height. One of the main differences between the two methods was that FEM used a visual check on the results to manually correct points where the distance offset was too high.

### Classification

A large number of ecological, environmental, and conservation-oriented questions depend on species identification. This includes efforts to conserve individual species, understand and maintain biodiversity, and incorporate the biosphere into global circulation models (*Rocchini et al., 2015*; *Lees et al., 2018*). Species identification is

generally treated as a supervised problem, whose demand for labelled data is usually high. Linking remote sensing with field data would potentially provide species identifications for thousands of trees, facilitating the building of more successful classifiers. For this reason, we identified species classification as the last step of the pipeline (Fig. 1). Classifying trees species from remote sensing imagery is complicated by: (1) highly unbalanced data, where some species are more abundant than others, so the amount of data for different species may differ orders of magnitude; (2) features fundamental to differentiating species that cannot be perceived by the human eye; (3) contribution of the understory and soil to the image properties for ITCs; and (4) data limitation, especially for rare species. A detailed description of the state of the art can be found in *Fassnacht et al. (2016)*, and other methods borrowed by the field of Image Vision as described in *Wäldchen & Mäder (2018)*.

*Performance metric.* We evaluated classification performance using two metrics. The first was rank-1 accuracy, namely the fraction of crowns in the test set whose ground truth species identification (species_id) and genus identification (genus_id) was assigned the highest probability by the participant. It is calculated as:

$$\text{rank}_1 = \frac{\sum_{n-1}^{N} \text{argmax}_k(p_{nk}) == g_n}{N}$$

where $g_n$ is the ground-truth class of crown $i$, and $p_{nk}$ is the probability assigned by the participant that crown $i$ belongs to class $k$. This metric only considers whether the correct class has the highest probability, not whether the probabilities are well-calibrated.

The second metric was the average categorical cross-entropy, defined as:

$$\text{cost} = \frac{-\sum_{n,k}^{N} \ln(p_{nk})\delta(g_n, k)}{N}$$

given that $p_{nk} \neq 0$, to avoid the singularity. The $\delta(x, y)$ is an indicator function that takes value 1 when $x = y$. This metric rewards participants for submitting well-calibrated probabilities that accurately reflect their uncertainty about which crowns belong to which class.

*Algorithms.* Our baseline prediction was a classification based on probability distributions of species frequency in the training data. We chose this baseline because it is a null model which uses no information from the features. The Conor group reduced the first 10 components of the hyperspectral data and CHM information to three components, with two principal component analysis subsequently (*McMahon, 2018*). They applied a maximum likelihood classifier to the test set to calculate the probability of each test tree to be a specific tree of the training set. The class (species) of the tree in the test set was assigned by using the same label of the individual tree with highest likelihood. The BRG group used a neural network multi-layer perceptron on the hyperspectral images (*Sumsion et al., 2018*). Crown probabilities were aggregated by averaging the pixel scale predicted probabilities. FEM applied a four step pipeline, consisting of data normalization, sequential forward floating feature selection, building of a support vector machine classifier, and crown level aggregation by majority rule
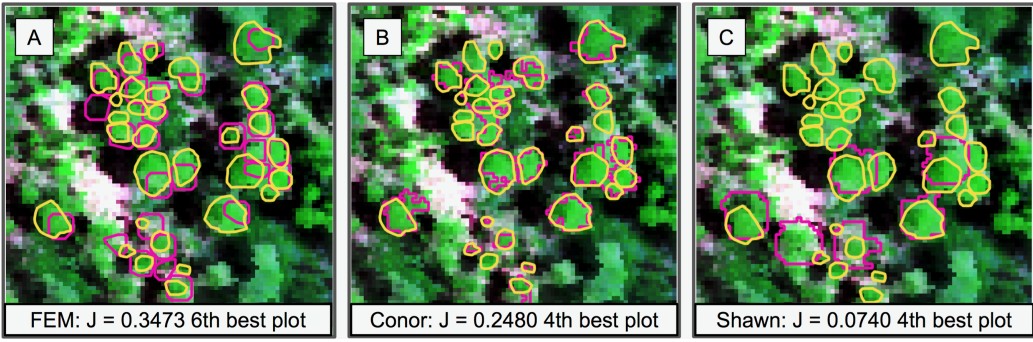

**Figure 2 Sample of the participants' algorithm performance on plot 41, ranked around the median in performance for all the three groups (ranking sixth, fourth, and fourth, respectively).** Plot 41 represented the median plot in terms of average performance for all the three groups. J is the average Jaccard index for that plot. (A) It ranked sixth for group FEM from the *Fondazione Edmund Mach*. (B) It ranked fourth for group Conor from the University of Texas at Austin. (C) It ranked fourth for group Shawn from the University of Florida. Yellow polygons represent ground truth ITCs, magenta the predicted ITCs. The background image is a composite of the hyperspectral data (R = 829 nm, G = 669 nm, B = 473 nm).

(*Dalponte, Frizzera & Gianelle, 2018*). The GatorSense group built a series of one-vs-one applied multiple instance adaptive cosine estimator classifiers (*Zare, Jiao & Glenn, 2017*; *Zou, Gader & Zare, 2018*) that automatically select the best subset of pixels to use for classification. Crown level probabilities were assigned by majority vote of pixel scale predictions. Finally, StanfordCCB group applied a six step pipeline (*Anderson, 2018*). Dimensionality reduction was performed using principal components analysis, and the first 100 components were retained. Pixels with high shade fractions were removed. Random Forest and Gradient Boosting multi-label classification algorithms were applied in a one-vs-all framework. Training species were under- or over-sampled to deal with label imbalance. Hyperparameters were determined using a grid search function, and prediction probabilities were calibrated using validation data. Finally, prediction probabilities were averaged between the two model ensembles.

## RESULTS

Overall, there was no single team that had a highest performing system across all three tasks. The FEM group achieved the highest evaluation scores for the segmentation and alignment tasks, but had a lower score for the classification task than the highest scoring group, StanfordCCB. In all three tasks, the highest scoring group scored substantially higher than the baseline. Given our evaluation data and metrics for each task, some groups performed better than the others. However, there is useful information in the approaches of teams that did not achieve the best performance on this specific competition configuration.

### Segmentation

This task had the lowest performance among the three tasks given our evaluation data and criteria (Fig. 2). A segmentation that perfectly matched our field-delineated crowns would achieve of Jaccard score of 1.0000. All submissions performed well below the

**Table 4 Comparison of Jaccard scores among submissions and baseline.**

**Task 1: Crown delineation**

| Rank | Participant | Score |
| --- | --- | --- |
| #1 | FEM | 0.3402 |
| #2 | Conor | 0.184 |
| #3 | Shawn | 0.0555 |
| | Baseline | 0.0863 |

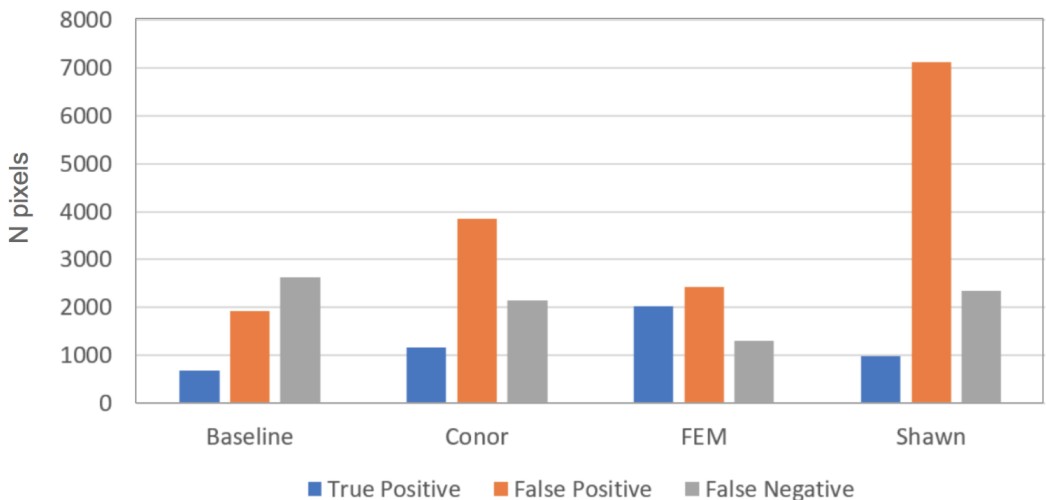

**Figure 3 Summary of error types for the crown segmentation task, using the 2 by 2 confusion matrix.**
The bar plot shows number of true positive (blue), false positive (orange), and false negative (gray) pixels for the different methods evaluated in task 1 (segmentation). Although presented in this figure, we did not use false negatives in the current competition evaluation criteria, since the ground truth ITCs did not cover the entire image area. For this reason, the number of pixels obtained by summing the three columns per each group do not necessarily match among submissions. Baseline shows results for the "Chase-Vese" algorithm, the Conor group is from the University of Texas at Austin, FEM is from the Fondazione Edmund Mach, and Shawn is from the University of Florida.

optimal score, but well above the baseline prediction. The highest-performing method, as determined by the Jaccard scoring function, achieved score of 0.3402 (Table 4). In comparison, our baseline system only has a score of 0.0863. All groups had more false positives compared to true positives, suggesting that all groups made polygons bigger than the field-based ITCs, on average (Fig. 3). Only two groups, baseline and Connor (*McMahon, 2018*), had more false negatives than true positives indicating these approaches failed to segment some portion or all of a crown. Overall, the FEM group (*Dalponte, Frizzera & Gianelle, 2018*) had the best balance between minimizing false positive and negatives as well as the highest number of true positives, across trees with different crown size (Fig. 4).

## Alignment

In this task, the FEM group again achieved the best performance, while the baseline system and the Conor group performed equally well. Surprisingly, the FEM group had the perfect accuracy score of 1.0 (Fig. 5). However, their pipeline is not fully automatable,

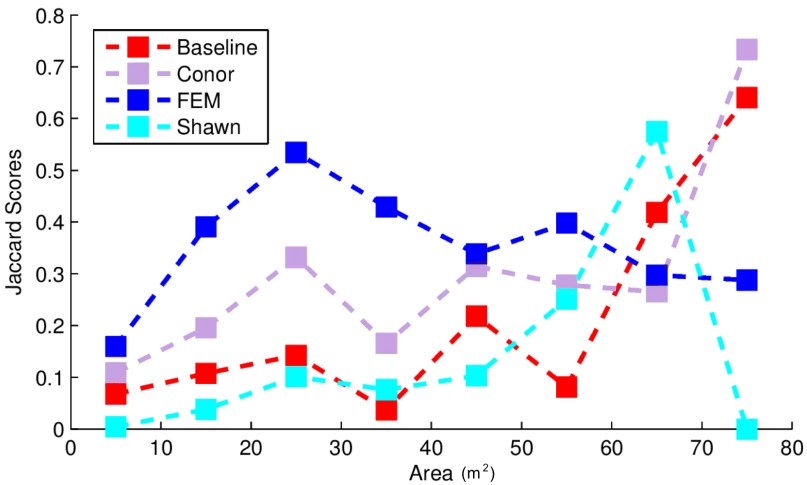

**Figure 4 Jaccard score for crown segmentation as a function of the size (area) of the tree crown.** Jaccard scores for individual trees are binned into size classes and averaged. Baseline (Red) shows results for the "Chase-Vese" algorithm, the Conor group is from the University of Texas at Austin (Green), the FEM is from the Fondazione Edmund Mach (Blue), and Shawn from the University of Florida (Azure).

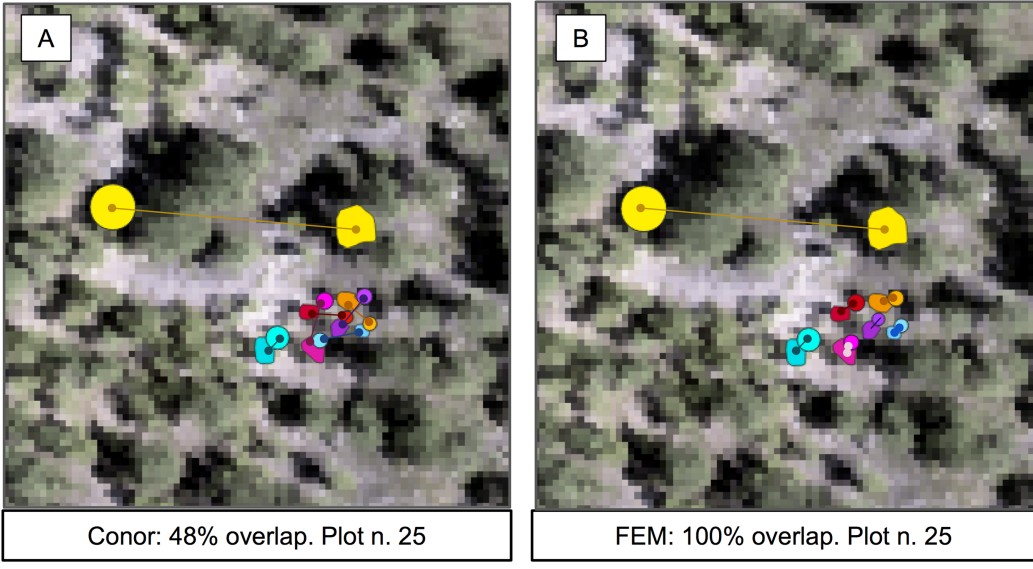

**Figure 5 Sample of the participants' algorithm performance on plot 25, for the two competing groups.** (A) Performance on the alignment task for group FEM from the *Fondazione Edmund Mach.* (B) Performance on the alignment task for group Conor from the University of Texas at Austin. Plot 25 was chosen for visualization because of the presence of one highly misaligned crown. Data shown are stem locations (circles) scaled by diameter at breast height; field ITCs (polygons); euclidean distances between the two data sources with same identity (solid line). ITCs, stem, and distances colored by stem identity. The background image is a desaturated hyperspectral composite (R = 829 nm, G = 669 nm, B = 473 nm).

and so may not be fully reproducible or scale to a significantly larger spatial extent. On the other hand, despite the similar structure to the automated part of FEM's method, the Conor group did not perform any better than the baseline (Table 5).

**Table 5 Comparison of alignment accuracy among submissions and baseline.**

**Task 2: Crown alignment**

| Rank | Participant | Score |
|------|-------------|-------|
| #1 | FEM | 1 |
| #2 | Conor | 0.48 |
| | Baseline | 0.48 |

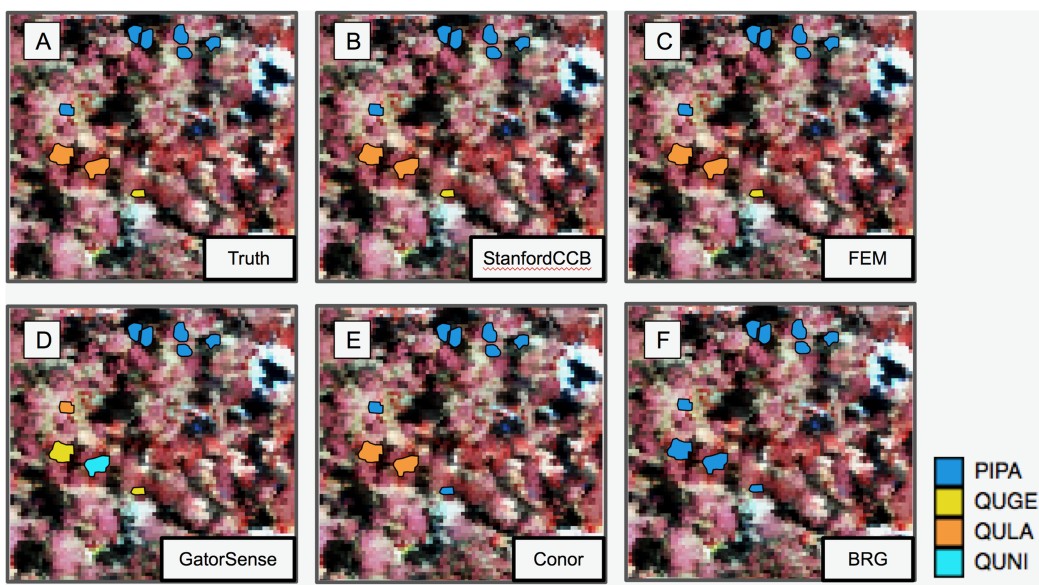

**Figure 6 Performance of species classification in a plot that is relatively diverse in species composition.** Field ITCs colored by species code for (A) ground observations. (B) StanfordCCB from Stanford University. (C) FEM from the Fondazione Edmund Mach. (D) GatorSense from the University of Florida. (E) Conor from the University of Texas at Austin. (F) BRG group from Brigham Young University (Red). The background is a false-color composite of the hyperspectral data (R = 829 nm, G = 549 nm, B = 473 nm). Species shown in the legend are: *Pinus palustris* (PIPA), *Quercus geminata* (QUGE), *Quercus laevis* (QULA), and *Quercus nigra* (QUNI).

## Classification

We had the most participants in this task (six): BRG (*Sumsion et al., 2018*), Conor (*McMahon, 2018*), FEM (*Dalponte, Frizzera & Gianelle, 2018*), GatorSense (*Zou, Gader & Zare, 2018*), StanforCCB (*Anderson, 2018*), and our baseline system (Fig. 6). For the evaluation criteria used in this competition, cross entropy loss (CE) and Rank-1 accuracy (Rank1), there was consistent ranking of all groups except our baseline system (Table 6). The top three groups in order were StanfordCCB, FEM, and Gatorsense (Fig. 7). Conor and BRG outperformed our baseline system in Rank1 but not CE. Most of the difference in accuracy among groups was determined by ability in classifying species that were infrequent in the data set. In fact, all groups performed well in predicting the two most common species *P. palustris* and *Q. laevis*, according to Rank1 scores (Fig. 8). However, the three lowest-performing approaches (Baseline, BRG, and Conor) failed to

**Table 6 Comparison of classification performance on categorical cross-entropy and rank-1 accuracy among submissions and baseline.**

| Rank | Participant | Score (cross entropy) | Score (Rank-1 accuracy) |
|------|-------------|-----------------------|--------------------------|
| #1 | StanfordCCB | 0.4465 | 0.9194 |
| #2 | FEM | 0.8769 | 0.88 |
| #3 | GatorSense | 0.9386 | 0.864 |
| #4 | Conor | 1.2247 | 0.8226 |
| #5 | BRG | 1.4478 | 0.688 |
| | Baseline | 1.1306 | 0.6667 |

Task 3: Species classification

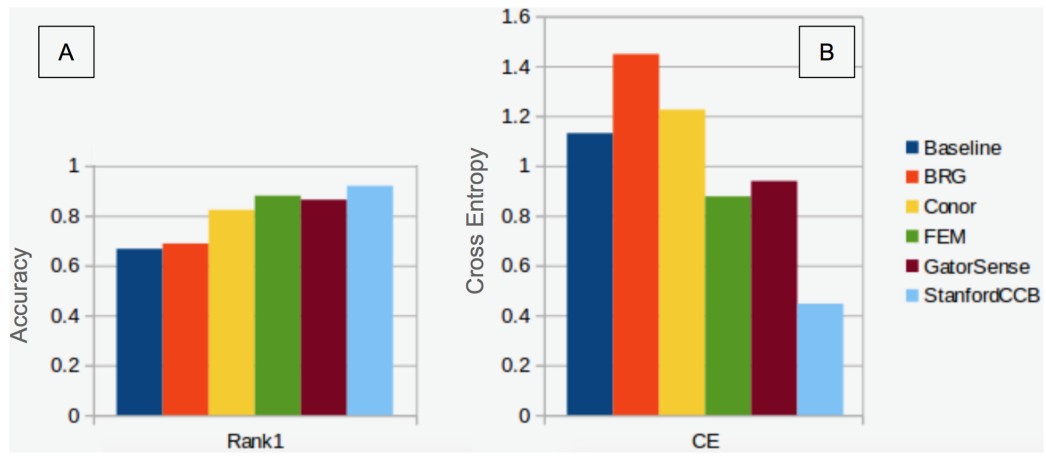

**Figure 7 Classification performance comparison.** (A) Rank 1 accuracy (from 0 to 1, the higher the values, the more accurate the method). (B) Categorical cross-entropy (from 0 to infinity, the lower the value the more accurate the model). Colors and labels refer to the six algorithms evaluated in this task. Baseline algorithm (Blue), BRG group from Brigham Young University (Red), Conor from the University of Texas at Austin (Yellow), FEM from the Fondazione Edmund Mach (Green), GatorSense from the University of Florida (Brown), and StanfordCCB from Stanford University (Azure).

predict all but these two species. StanfordCCB, FEM, and GatorSense were able to predict both PIPA and the rarest species (i.e., LIST and QUNI), but performed differently for the other species.

## DISCUSSION

The results of the competition are both promising and humbling, and the results for each task provide different lessons for how to improve both the conversion of remote sensing to ecological information, and the competition itself. An assessment of the results for each of the individual tasks is provided below.

### Crown segmentation

The results of the crown segmentation task reveal the challenging nature of segmentation problems (*Zhen, Quackenbush & Zhang, 2016*). The highest-performing algorithms
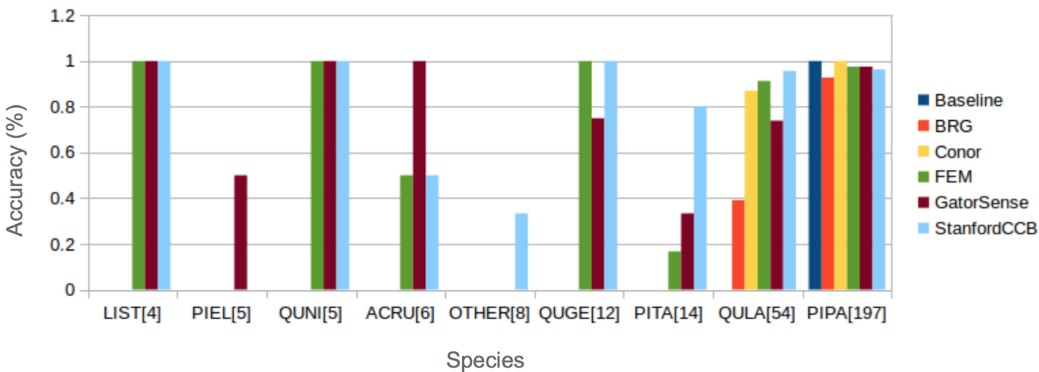

**Figure 8 Comparison of Rank-1 classification accuracy by species.** The number in square bracket is number of training samples. Baseline algorithm (Blue), BRG group from Brigham Young University (Red), Conor from the University of Texas at Austin (Yellow), FEM from the Fondazione Edmund Mach (Green), GatorSense from the University of Florida (Brown), and StanfordCCB from Stanford University (Azure). For cases when accuracy is 0, bars are missing. Species shown in the legend are: *Liquidambar styraciflua* (LIST), *Pinus elliottii* (PIEL), *Quercus nigra* (QUNI), *Acer rubrum* (ACRU), *Unknown* (OTHER), *Quercus geminata* (QUGE), *Pinus taeda* (PITA), *Quercus laevis* (QULA), *Pinus palustris* (PIPA).

yielded only 34% overlap between the closest remotely sensed crowns and ground truth crowns mapped directly onto remote sensing imagery in the field. This suggests that crown segmentation algorithms have substantial room for improvement for precisely identifying individual crowns from remote sensing imagery.

By looking at the results across the three algorithms for this task, we can identify future directions for improvement. FEM, the best performing method, was the only method using hyperspectral data to perform segmentation, despite LiDAR data being used more commonly for segmentation (*Zhen, Quackenbush & Zhang, 2016*). This indicates that there is useful information in the hyperspectral data for classification. For example, the hyperspectral data may allow distinguishing overlapping crowns from different species. As a result, some participants suggested that better segmentation may be achieved in the future by combining both hyperspectral and LiDAR derived information (*McMahon, 2018*; *Dalponte, Frizzera & Gianelle, 2018*). However, it should be noted that the ground truth polygons were identified using the hyperspectral data (and not the LiDAR). This means that any misalignment resulting from preprocessing and orthorectification of the hyperspectral and LiDAR data would advantage hyperspectral data over LiDAR for this task. NEON data showed one to two m misalignment in some of the images used for this competition. In fact, despite these two data products being co-registered, areas at the edges of an image can show misalignment between hyperspectral and LiDAR-based products, enough to drop in Jaccard Index score on predicted crowns significantly.

This source of uncertainty is important beyond this competition because LiDAR data is typically used to perform segmentation, while hyperspectral data is usually used for classification. In case of misalignment, the exact segmentation on LiDAR would result in imperfect inclusion of hyperspectral pixels within associated crowns. As a result, LiDAR to hyperspectral misalignment should be taken into consideration when working with these data sources together and we will actively address it in future rounds of this DSE.

Exploring the accuracy of different segmentation algorithms more thoroughly reveals that uncertainty in delineating crowns is generally dependent on crown size (Fig. 4). This is important because depending on forest types, crowns size distribution may be very different. For example, crown area for 75% of the field-based ITCs ranged between 10 and 25 m$^2$. Crowns below 10 m$^2$ were poorly classified by all algorithms and most algorithms performed best for crown sizes over 40 m$^2$. This may be due to the fact that small crowns are often closer together, more heterogeneous in shape, and composed of fewer pixels. The highest-performing method, FEM's region growing algorithm, outperformed other algorithms on small and intermediate sized crowns. However, it performed worse than some other methods for the largest crowns. Conor's and Shawn's methods generally performed best for larger crowns. This result shows the value of a comparative evaluation of different families of methods and suggests that creating ensembles of existing algorithms could result in better crown segmentation across the full range of tree sizes.

## Alignment

The results for the alignment tasks were promising. In fact, FEM's Euclidean distance based approach produced a perfect alignment between remotely sensed crowns and the stem location of individual trees. This precise match was accomplished by considering not only the position of the stem, but also the size of the crown. Adding the size of the crown was crucial for successful alignment because it allowed the algorithm to differentiate between multiple nearby stems based on differences in size. Using only Euclidean distance based on the position of the stems (the baseline) resulted in only a 48% alignment between stems and crowns. This perfect alignment is particularly encouraging because it used a statistical relationship between a standard field based measure of tree size (height) to estimate the size of the crown for the field data in cases where crown size was not measured. This means that the approach can be applied to all trees measured in the field, not just those where the less common direct measures of crown dimensions are performed. However, it is worth noting that FEM also performed a visual check of the alignments and shifted a few alignments manually based on this assessment (*Dalponte, Frizzera & Gianelle, 2018*). This yielded meaningful improvements for crowns with misalignments of several meters or more (likely resulting from data entry or collection errors). While including manual steps is typically a concern for scaling up remote sensing predictions, it is less of an issue for alignment since this step is only important for model building, not prediction. That means that this step will typically only be applied to a few hundred or thousand trees making human involvement doable and potentially important.

While the alignment results are encouraging for linking remote sensing and ground truth data at the individual level, in hindsight, the degree of this success was also due in part to how we posed the problem for the competition. When selecting data for this task we only included trees that occurred in both the field and remote sensing data. In all cases, there were additional trees in the 80 x 80 m image subsets that were not included in both the field and remote sensing data. This simplification resulted in overly sparse data compared to real-world situations where field data would need to be aligned against a full scene of remotely sensed crowns. Our original decisions made sense from an

assessment perspective but failed to reflect the real-world complexity of the problem. We expect that including all trees in the scene will make the task more challenging. In the next round of the competition, we plan to include the remotely sensed crowns that lack corresponding field data to provide a clearer picture of the effectiveness in real-world situations.

## Classification

The species classification task was led by the StanfordCCB algorithm, which yielded the best overall performance with a categorical cross-entropy of 0.45 and a rank-1 accuracy of 92% (Fig. 7). This is on the high end of classification accuracy rates reported for tree species identification from remote sensing (*Fassnacht et al., 2016*). This approach involved multiple preprocessing steps and an ensemble of random forest and gradient boosting multi-label classifications applied on each tree in a one-vs-all framework. A number of different models also performed well with rank-1 accuracies greater than 80% including Gatorsense, FEM, and Conor. StanfordCCB performed better in relation to other models when evaluated using categorical cross-entropy compared to rank-1 accuracy, which suggests that this method provides more accurate characterizations of uncertainty. Therefore, it is good at both identifying which species class a tree is most likely to belong to, and at knowing when it is unsure of which species to predict. This is a desirable property for a remote sensing model because good estimates of uncertainty allowing accurate error propagation into applications of those models. Exploring these results further by evaluating classifications for individual species (Fig. 8; not part of the defined goals of the competition) shows that the StanfordCCB, FEM, and Gatorsense methods provide the best classifications for rare species, while other methods are only accurate for common species.

Interestingly, most of the groups that performed well developed multi-step methods that used data cleaning and dimensionality reduction. Outlier removal such as filtering dark or non-green pixels, seemed to be particularly important, likely because it allowed shadowed pixels or pixels mixed with non-green vegetation like soil and wood, to be removed from the analysis. The Conor group averaged the spectra across all crown pixels and used structural information, namely crown radius and height range. Interestingly, averaging crown spectral information resulted in high predictability of the two most dominant classes, yet it was not a good strategy to predict rare species. This result suggests that clearing mixed noisy pixels may be particularly effective to better predict rare species. Likewise, adding structural features like crown radius may be useful in separating dominant classes. In general, the groups which performed best involved people with ecological expertise, which appeared useful in processing and selecting meaningful features from the data.

The other interesting aspect of the third task was the high participation. Five teams participated in this task compared to two teams for task 1 and three teams for task 2. We suspect that the higher level of participation was due to the task being the most straightforward, out-of-the-box, analysis. The relevant data was already extracted into a common tabular form meaning that most classification algorithms could be applied

directly to the provided data. This makes the task easier for non-domain experts and suggests that standardizing tasks, so that a common set of algorithms can be readily applied to them, could result in greater participation in this type of competition and result in broad improvements across disciplines. This is the motivation behind a new NIST effort focused on algorithm transferability where the goal is to allow algorithms developed in one field to be applied to similar problems in other disciplines. The next iteration of the NIST DSE Series (*Dorr et al., 2016a*, *2016b*) will combine sets of related tasks from different domains to help drive this idea of algorithm transferability forward. Accomplishing this requires standardizing data formats to allow integration into a central automatic-scoring system. We are in the process of converting the data from this competition into schema provided by DARPA's data-driven discovery of models program (D3M) for this purpose.

Dealing with complex and non-standard data types also highlights some of the challenges for data competitions in the environmental sciences. For example, most of the data in this competition is spatially explicit, a data type that does not completely generalize to more standard, non-spatial contexts, and involves file formats that many potential participants are not familiar with. We mitigated some of these challenges by cleaning and extracting simpler products from the data. For example, we provided the hyperspectral and CHM information in a single csv file, rather than as a collection of shapefiles and images. However, this strategy also resulted in a loss of information relevant to some tasks and algorithms. As an example, one participant found that the choices we had made to simplify the data prevented their use of more advanced tools (like convolutional neural networks) on the classification task. In future rounds, we will seek to both provide simplified representations of the data that are accessible to many users and the full raw data that allow experts to employ tools appropriate to that data type.

## Insights from the competition

We developed and ran a data science competition on converting airborne remote sensing data into information on individual trees, with the goal of improving methods for using remote sensing to produce ecological information and accelerating development of methods in ecology more broadly. In developing this competition we took advantage of a major new source for open ecological and remote sensing data, the NEON. Because of the long-term large-scale nature of NEON's data collection, the results of the competition have the potential to go beyond general improvements in methods to yield immediate improvements in the quality of the ecological information that can be extracted from this massive data collection effort. The clearly defined goals, potential for general methodological improvements, and opportunity for immediate operationalization to produce data products that will be used by large numbers of scientists, makes this an ideal combination of problems and data for a data science competition.

We identified a single algorithm for each task that had the highest performance based on one or two performance criteria. While these algorithms showed the greatest promise for maximizing the evaluation criteria—for example, providing the highest rank-1 classification accuracy for species identification—caution should be taken in focusing

too much on a single method for several reasons. First, there are many different evaluation criteria that can be used depending on the specific application and ecological questions to be addressed. For example, in the evaluation criteria for the classification task, the correct identification of all trees was weighted equally, such that an algorithm that could correctly predict the common species would be favored over an algorithm that correctly predicted the rare species. Correct identification of the most common species may be the key goal for some ecological questions, such as producing maps of aboveground biomass. On the other hand, there may be other ecological questions for which equally good classification for all species is desirable. In this case, the training and test data may be chosen so that it is balanced among species, or weighting used in the evaluation criteria to increase the importance of identifying less common species (*Graves et al., 2016*; *Anderson, 2018* did this for this competition). For some biodiversity assessments, the optimization for the species classification task may be more focused on identifying rare species, a single exotic species, or identifying species that are outliers, and potentially "new" or unusual species in the system (*Baldeck et al., 2015*). The evaluation criteria for these alternative goals would differ from the ones used in this competition.

In addition to performing differently for a variety of specific tasks, different algorithms may vary in applicability and performance in different ecosystems or when using different types of field data. This competition used ITCs from forest ecosystems where the average crown size was relatively small (around 20 m$^2$). According to Fig. 4, in other forests such as western oak savannas, with big isolated crowns of sizes greater than 40 m$^2$, methods using CHM (Conor and Baseline) may perform better at crown segmentation than those using hyperspectral data (FEM). Moreover, in forest ecosystems, the unit of observation are multi-pixel tree objects. However, other ecosystems, such as grasslands, prairies and shrublands, are dominated by plant species whose size is below the resolution of an individual pixel. NEON provides extensive data sets on the presence and cover of small plant species that if linked with NEON–AOP data, could be used to generate landscape maps of these species. At a number of sites these sub-pixel plant species are the dominant plants at the site. Working across all NEON sites will therefore require algorithms that can perform alignment and identification of both super- and sub-pixel resolutions, a complex task that may change which algorithms perform best. For example, one of the approaches used in this competition, GatorSense's multiple instance classification, had slightly lower performance than the highest-performing method in the species classification task, but has the flexibility to be used for the alignment and subpixel detection of small plant species presence and cover (*Zare, Jiao & Glenn, 2017*). This suggests that despite not being the highest-performing method in the competition, it is a promising route forward for the more general task.

Moreover, our findings apply to data collected at a spatial resolution that provides several pixels per individual crown. Other remote sensing datasets provide different spatial resolutions, a different number of spectral bands and different spectral regions. Applying the same methods on other data sources could yield different evaluation ranking of these methods. In fact, while we expect similar rankings from data collected by
NASA G-LiHT (https://gliht.gsfc.nasa.gov) because of the similar spatial and spectral resolution, that may not be true for AVIRIS-NG data (https://aviris-ng.jpl.nasa.gov). AVIRIS-NG data provides similar spectral resolution but coarser pixel size (between 16 and 25 m$^2$), often resulting in having a single pixel per one or more individual crowns. Coarser spatial resolution mean methods heavily relying on hard classifiers work more poorly, and favor soft classifiers or algorithms based on spectral unmixing. Future competitions could include different data sources to investigate how proposed methods perform with image data with different spectral and spatial resolutions.

For competitions like this one to be most effective in facilitating rapid methodological improvement of a field, it is important that the details of each teams' analysis be described in detail and easy to reproduce. This allows for researchers to quickly integrate the advances made by other participants into their own workflows. We accomplished this for this competition in three ways. First, all of the data is openly available under an open license (*ECODSE group, 2017*). Second, all authors wrote short papers describing the detailed methods employed in their analyses and these papers are published as part of collection associated with this paper (https://peerj.com/collections/56-remotesensingcomp/). Finally, all authors posted their code openly on GitHub and linked it in their contributions. One author (*Anderson, 2018*) even encouraged other researchers to use and further improve on their method with the hope of collaboratively improving the use of remote sensing for species classification. Having access to a growing number of fully reproducible open pipelines evaluated on the same data will be a powerful instrument improving the methods used in converting remote sensing into ecological information.

We plan to continue to run this competition, updating the specifics of the tasks to help advance the science of converting remote sensing to information on individual trees. In the next iteration of this competition, we plan to address the fact that remote sensing models for identification of species and other key ecosystem traits are usually developed at individual sites (*Zhen, Quackenbush & Zhang, 2016*; *Fassnacht et al., 2016*), which tend to make them site-specific and leads to a profusion of locally optimized methods that do not transfer well to other locations. For standardized data collection efforts like NEON, algorithms and models that perform well across sites are critical. To facilitate advances in this area we will include data from multiple NEON sites in future competitions with the goals of developing algorithms with high cross-site performance and comparing the performance of cross-site and site-specific algorithms.

## CONCLUSIONS

The results of this competition are encouraging both for the specific scientific tasks involved and for the use of competitions in ecology and science more broadly. The highest performing algorithms are indicative of the potential for using remote sensing models to obtain reasonable estimates of the location and species identity of individual trees. The competition results help highlight the components of this process that have good existing solutions as well as those most in need of improvement. Promising areas for future development include the ensemble of crown segmentation algorithms that perform well for small vs large crowns. In cases with clearly defined outcomes, science would

benefit from the increased use of competitions as a way to quickly determine and improve on the highest-performing methods currently available.

## ACKNOWLEDGEMENTS

The NEON is a program sponsored by the NSF and operated under cooperative agreement by Battelle Memorial Institute. This material is based in part upon work supported by the NSF through the NEON Program. These results are not to be construed or represented as endorsements of any participants system, methods, or commercial product, or as official findings on the part of NIST or the U.S. Government. Certain commercial equipment, instruments, software, or materials are identified in this paper in order to specify the experimental procedure adequately. Such identification is not intended to imply recommendation or endorsement by NIST, nor is it intended to imply that the equipment, instruments, software or materials are necessarily the best available for the purpose.

### Funding

This research competition was supported, in part, by a research grant from NIST IAD Data Science Research Program, by the Gordon and Betty Moore Foundation's Data-Driven Discovery Initiative through grant GBMF4563 to Ethan P. White, and by the NSF Dimension of Biodiversity program grant (DEB-1442280) and USDA/NIFA McIntire-Stennis program (FLA-FOR-005470) to Stephanie Ann Bohlman. This work was supported in part by a University of Florida Biodiversity Institute Graduate Fellowship to Sergio Marconi. There was no additional external funding received for this study. As is standard for government agencies the paper underwent internal review by NIST. Comments from these four authors and that review were incorporated into the manuscript, but focused on communication of ideas not interpretation of results.

### Grant Disclosures

The following grant information was disclosed by the authors:
NIST IAD Data Science Research Program, by the Gordon and Betty Moore Foundation's Data-Driven Discovery Initiative: GBMF4563.
NSF Dimension of Biodiversity program: DEB-1442280.
USDA/NIFA McIntire-Stennis program: FLA-FOR-005470.

### Competing Interests

Marion Le Bras, Bonnie J. Dorr, Peter Fontana, and Craig Greenberg are employee of the National Institute of Standards and Technology (NIST), which funded this study.

As is standard for government agencies, the paper underwent internal review by NIST. Comments from these four authors and that review were incorporated into the manuscript, but focused on communication of ideas not interpretation of results.

Ethan P. White is an Academic Editor for PeerJ.

## Author Contributions

- Sergio Marconi conceived and designed the experiments, performed the experiments, analyzed the data, contributed reagents/materials/analysis tools, prepared figures and/or tables, authored or reviewed drafts of the paper, approved the final draft.
- Sarah J. Graves conceived and designed the experiments, performed the experiments, analyzed the data, contributed reagents/materials/analysis tools, prepared figures and/or tables, authored or reviewed drafts of the paper, approved the final draft.
- Dihong Gong conceived and designed the experiments, performed the experiments, analyzed the data, contributed reagents/materials/analysis tools, prepared figures and/or tables, authored or reviewed drafts of the paper, approved the final draft.
- Morteza Shahriari Nia conceived and designed the experiments, performed the experiments, contributed reagents/materials/analysis tools, authored or reviewed drafts of the paper, approved the final draft.
- Marion Le Bras contributed reagents/materials/analysis tools, authored or reviewed drafts of the paper, approved the final draft.
- Bonnie J. Dorr contributed reagents/materials/analysis tools, authored or reviewed drafts of the paper, approved the final draft.
- Peter Fontana contributed reagents/materials/analysis tools, authored or reviewed drafts of the paper, approved the final draft.
- Justin Gearhart contributed reagents/materials/analysis tools, authored or reviewed drafts of the paper, approved the final draft.
- Craig Greenberg contributed reagents/materials/analysis tools, authored or reviewed drafts of the paper, approved the final draft.
- Dave J. Harris conceived and designed the experiments, contributed reagents/materials/analysis tools, authored or reviewed drafts of the paper, approved the final draft.
- Sugumar Arvind Kumar performed the experiments, contributed reagents/materials/analysis tools, authored or reviewed drafts of the paper, approved the final draft.
- Agarwal Nishant performed the experiments, contributed reagents/materials/analysis tools, authored or reviewed drafts of the paper, approved the final draft.
- Joshi Prarabdh performed the experiments, contributed reagents/materials/analysis tools, authored or reviewed drafts of the paper, approved the final draft.
- Sundeep U. Rege performed the experiments, contributed reagents/materials/analysis tools, authored or reviewed drafts of the paper, approved the final draft.
- Stephanie Ann Bohlman conceived and designed the experiments, performed the experiments, analyzed the data, contributed reagents/materials/analysis tools, prepared figures and/or tables, authored or reviewed drafts of the paper, approved the final draft.
- Ethan P. White conceived and designed the experiments, performed the experiments, analyzed the data, contributed reagents/materials/analysis tools, prepared figures and/or tables, authored or reviewed drafts of the paper, approved the final draft.
- Daisy Zhe Wang conceived and designed the experiments, performed the experiments, analyzed the data, contributed reagents/materials/analysis tools, prepared figures and/or tables, authored or reviewed drafts of the paper, approved the final draft.

## Data Availability

ECODSE group. (2017). ECODSE competition training set [Data set]. Zenodo. http://doi.org/10.5281/zenodo.1206101.

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
