# Peer review of "A data science challenge for converting airborne remote sensing data into ecological information"

_PeerJ, doi:10.7717/peerj.5843_

## Round 0.1 · original submission · Minor Revisions

This manuscript presents an overview of a data science competition in the application of remotely-sensed data to ecology. I found the concept and the paper itself very appealing. The reviewers and I had only a few concerns that should be addressed in revisions.

1. Please address in more detail the applicability of these methods (and the specific ranking of methods) to other hyperspectral datasets and study locations or ecosystems

2. Discuss the implications of the temporal mismatch between the collection of the hyperspectral data and the ground-truth crown-delineation data.

3. Please address the comments on the figures and captions raised by Reviewer 1.

Reviewer 1 ·

Basic reporting

The manuscript is concise and clear. The authors provided a good background for the research topics addressed in the manuscript: the data science competition (a concept that I found very interesting) and processing of remotely sensed images to extract tree crown size and location, align images with field reference data (ground truth), and identify individual trees. The figures illustrate well the methods and results of the study. A few editing suggestions for figure captions, to make them more descriptive:
Figure 1 caption: specify which bands (wavelengths) were used create the false-color composite; add the names of the species used in the legend (species codes);
Figure 1 design: why does panel C include three superimposed images?
Figure 2 caption: rephrase “ranked around the median highest in performance”; specify which bands (wavelengths) were used create the composite; include information about the three groups (FEM, Conor, Shawn)
Figure 3 caption: remove the comma after “criteria”; add information about the three groups (FEM, Conor, Shawn) used on X axis
Figure 4 caption: include information about the three groups (FEM, Conor, Shawn)
Figure 4 design: add units on X axis
Figure 5 caption: specify which bands were used create the hyperspectral composite image
Figure 6 caption: include names of species presented in legend as species codes and provide information about the labels used on the images (groups); specify which bands were used create the false-color composite
Figure 7 caption: provide information about the labels used in the figure legend
Figure 8 caption: provide information about the labels used in the figure legend and explain the codes used on X axis (species’ names).
The formatting of tables could also be adjusted to resemble lists (remove vertical and horizontal lines, except for the horizontal line of the header and the end of the table).
Data used in this study are available through NEON portal.

Experimental design

The data science competition was well designed and the analysis of results obtained by various teams seems robust to me. I think that this approach of providing several teams with same data and analyzing the results obtained by teams individually is a good way to compare methods because it pools skills; each team will most likely choose the method(s) that the team has most knowledge of or is most interested in developing.
The methods used to compare the teams’ results are described in detail. The individual team’s methods are not included, but the papers of the teams are cited in this manuscript. It might be useful to add a table that summarizes each team’s methods, capturing just the essential characteristics.

Validity of the findings

The authors discussed the most important aspects of their research but also the possible limitations. One aspect that could be added to the discussion of this study is that the findings apply to NEON AOP hyperspectral data – other hyperspectral datasets (e.g., AVIRIS) might not yield the same conclusions. I think this aspect might be important to mention so that readers are reminded that the ranking of methods in this study may not apply to other hyperspectral data.

Additional comments

I enjoyed reading this manuscript. My review is structured according to the PeerJ format. Additional minor editing suggestions:
Lines 223-224: “widely” is repeated here
Line 240: provide citation for the Hungarian algorithm
Line 317: explain “highly unbalanced data”
Line 584: Anderson 2018 is missing from References list

Reviewer 2 ·

Basic reporting

The manuscript describes the results of a data competition organized in 2017 to achieve three important tasks essential for making remote sensing data useful for ecological applications - tree crown delineation, tree stem location and tree species classification. The competition generated broader interest (mainly from US and Europe), albeit only small number of responses (total of six - only 2-3 groups participated in the Task 1 and Task 2 ). The significant outcomes of this effort include a series of manuscripts about algorithms, together with openly-available source codes and data. The manuscript is generally well-written and structured. The raw data is available online.

Experimental design

The competition is designed well with statistical metrics to compare the performance of various algorithms with the baseline. Most of algorithms are described in the accompanying submissions, which are pointed to in the manuscript. The baseline methods (for the three tasks) are however not described in sufficient details. The manuscript could benefit from information as to why particular methodology for baseline prediction was selected compared to other existing approaches.

Validity of the findings

My one concern with the manuscript is the wider applicability of the findings - both training and test samples were from the same location (NEON Ordway-Swisher Biological Station in Florida). NEON already has similar data available for other stations/biomes. I feel that performance of the algorithms would have been tested more rigorously if the competition provided datasets from multiple sites, or at least the training and test data came from different NEON stations.

Given that most of the participants have also submitted the manuscripts at the PeerJ (McMahon et al., Sumsion et al., Dalponte et al., Anderson) or already published (Zare et al., 2017), this manuscript feel redundant and do not provide any additional insights that these individual manuscripts cannot provide. This is my main concern.

Additional comments

- In 'Introduction' section, the authors describe common data science competition platforms, specifically Kaggle, in detail. However, they fail to describe what platform they used to organize their own challenge. The authors should describe this and provide an URL to the competition website. The original URL to the competition is not provided anywhere in the manuscript.
- Hyperspectral imagery was collected in 2014 and baseline crown delineation field data (ITC) in 2017. The manuscript could benefit from additional discussion on the implications of this three years’ time lapse on the crown delineation, and hence the poor performance for Task 1 (?). Also mention in Section 2.2, which month or season the field data was collected.
- The authors mention possibility of misalignment between the LIDAR/Hyperspectral at several places in the manuscript, but do not state if they really found one. Weren’t these data already co-registered by NEON? Given that both were probably collected in the same flight, was there really a significant misalignment?
- Task1/Task3 in the Table 2 can be combined into one column. They essentially represent same values.
- Line 220: “LiDAR data” should be “LiDAR CHM data”. Can author also provide information on the distribution of the crown sizes?
- Line 240 – What is the “Hungarian algorithm”? Please describe.
- Section 2.5.1.2: Describe briefly the baseline algorithm (Chase-Vese algorithm”
- Lines 522-523 – Please state what exactly was done to transform the data
- Where is the baseline information for other species (other than PIPA) in Figure 8? State what the abbreviations mean (LIST, PIEL, QUNI, etc) in the figure caption.

---

## Round 0.2 · accepted · Accept

Thank you for your careful revision of the text and response to the reviewers. I look forward to seeing the paper in PeerJ.

#